# Neural Abstractions

**Alessandro Abate**[*]
Department of Computer Science
University of Oxford, UK

**Alec Edwards**[*]
Department of Computer Science
University of Oxford, UK

**Mirco Giacobbe**[*]
School of Computer Science
University of Birmingham, UK

## Abstract

We present a novel method for the safety verification of nonlinear dynamical models that uses neural networks to represent abstractions of their dynamics. Neural networks have extensively been used before as approximators; in this work, we make a step further and use them for the first time as abstractions. For a given dynamical model, our method synthesises a neural network that overapproximates its dynamics by ensuring an arbitrarily tight, formally certified bound on the approximation error. For this purpose, we employ a counterexample-guided inductive synthesis procedure. We show that this produces a neural ODE with non-deterministic disturbances that constitutes a formal abstraction of the concrete model under analysis. This guarantees a fundamental property: if the abstract model is safe, i.e., free from any initialised trajectory that reaches an undesirable state, then the concrete model is also safe. By using neural ODEs with ReLU activation functions as abstractions, we cast the safety verification problem for nonlinear dynamical models into that of hybrid automata with affine dynamics, which we verify using SpaceEx. We demonstrate that our approach performs comparably to the mature tool Flow* on existing benchmark nonlinear models. We additionally demonstrate and that it is effective on models that do not exhibit local Lipschitz continuity, which are out of reach to the existing technologies.

## 1 Introduction

Dynamical models describe processes that are ubiquitous in science and engineering. They are widely used to model the behaviour of cyber-physical system designs, whose correctness is crucial when they are deployed in safety-critical domains [10, 13, 47]. To guarantee that a dynamical model satisfies a safety specification, simulations are useful but insufficient because they are inherently non-exhaustive and they suffer from numerical errors, which may leave unsafe behaviours unidentified. Formal verification of continuous dynamical models tackles the question of determining with formal certainty whether every possible behavior of the model satisfies a safety specification [43, 49, 109]. In this paper, we present a method to combine machine learning and symbolic reasoning for a sound and effective safety verification of nonlinear dynamical models.

The formal verification problem for continuous-time and hybrid dynamical models is unsolvable in general and, even for models with linear dynamics, complete procedures are available under stringent conditions [11, 12, 70, 84, 85]. For most practical models that contain nonlinear terms [79, 103],

---

[*]The authors are listed alphabetically

36th Conference on Neural Information Processing Systems (NeurIPS 2022).

methods for formal verification with soundness guarantees involve laborious safety and reachability procedures whose efficacy can only be demonstrated in practice. Formal verification of nonlinear models require ingenuity, and has involved sophisticated analysis techniques such as mathematical relaxations [25, 32–34, 46, 101, 102], abstract interpretation [50, 51, 54, 80, 94], constraint solving [18, 37, 83], and discrete abstractions [5, 9, 28, 35]. Notwithstanding recent progress, both scalability and expressivity remain open challenges for nonlinear models: the largest model used in the annual competition has 7 variables [64]. In addition, existing formal approaches rely on symbolic reasoning techniques that explicitly leverage the structure of the dynamics. This results in verification procedures that are bespoke to restricted classes of models. For example, it is common for formal verification procedures to require the input model to be Lipschitz continuous. Yet, dynamical models with vector fields that violate this assumption are abundant in literature, and a wide variety of models of natural phenomena are non-Lipschitz, from fluid dynamics to n-body orbits and chaotic systems, as well as in engineering, from electrical circuits and hydrological systems [48, 53]. Our approach makes progress in expressivity, showing that using *neural networks as abstractions* of dynamical systems enables an effective formal verification of nonlinear dynamical models, including models that do not exhibit local Lipschitz continuity.

Abstraction is a standard process in formal verification that aims at translating the model under analysis—the concrete model—into a model that is simpler to analyse—the abstract model—such that verification results from the abstract model carry over to the concrete model [19, 38, 39]. In verification of systems with continuous time and space, an abstraction usually consists of a partitioning of the state space of the concrete model into a finite set of regions that define the states of an abstract, finite-state machine with a corresponding behaviour. Our method follows an approach that constructs abstract, finite-state machines whose states are augmented with continuous linear dynamics and non-deterministic drifts. Finite-state machines with continuous, possibly non-deterministic dynamics are known as hybrid automata [69], and the process of abstracting dynamical nonlinear models into hybrid automata is called *hybridisation*; this process has been widely applied in formal verification [8, 15, 16, 20, 44, 56, 63, 71, 89, 92, 97, 98, 100].

Hybridising involves partitioning the state space and computing a local *overapproximation* of the concrete model within each region of the partition. Common approaches for hybridisation partition the state space by tuning the granularity of rectangular or simplicial meshes, until a desired approximation error is attained. This may yield abstract hybrid automata that are too large in the number of discrete states to be effectively verified. Notably, modern tools for the verification of hybrid automata are designed for models that rarely have over hundred discrete states [7], while arbitrary meshes grow exponentially as the granularity increases. Explosion in discrete states has been mitigated using deductive approaches that construct an appropriate partitioning from the expressions that define the concrete model and, unlike our method, rely on syntactic restrictions [14, 24, 28, 45, 68, 78, 81, 82, 96].

We propose an *inductive approach to abstraction* that combines the tasks of partitioning the state space and overapproximating the dynamics into the single task of training a neural network. We leverage the approximation capability of neural networks with ReLU activation functions to partition the state space into arbitrary polyhedral regions, where each region and local affine approximation correspond to a combinatorial configuration of the neurons. We show that this ultimately enables verifying nonlinear dynamical models using efficient safety verifiers for hybrid automata with affine dynamics (cf. Figure 1).

Our abstraction procedure synthesises abstract models by alternating a *learner*, which proposes candidate abstractions, and a *certifier*, which formally assures (or disproves) their validity, in a counterexample-guided inductive synthesis (CEGIS) loop [106, 107]. First, the learner uses gradient descent to train a neural network that approximates the concrete model over a finite set of sample observations of its dynamics; then, the certifier uses satisfiability modulo theories (SMT) to check the validity of an upper-bound on the approximation error over the entire continuous domain of interest. If the latter disproves the bound, then it produces a counterexample which its added to the set of samples and the loop is repeated. If it certifies the bound, then the procedure returns a neural network approximation and a sound upper-bound on the error. Altogether, neural network and error bound define a neural ODE with bounded additive non-determinism that overapproximates the concrete model, which we call a *neural abstraction*.

We demonstrate the efficacy of our method over multiple dynamical models from a standard benchmark set for the verification of nonlinear systems [64], as well as additional locally non-Lipschitz

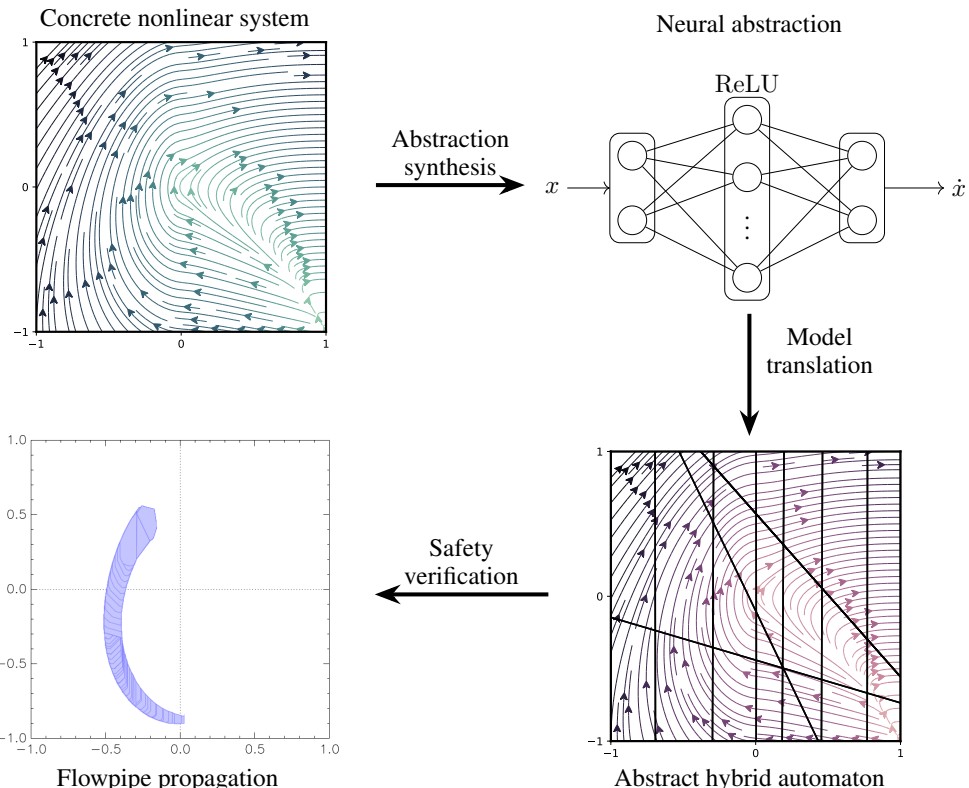

Figure 1: Overview of our workflow on a non-Lipschitz dynamical model (cf. Section 5, NL2). The concrete dynamics are abstracted by a neural ODE with ReLU activation functions and a certified upper-bound on the approximation error. This characterises a polyhedral partitioning and defines a hybrid automaton with affine dynamics and additive non-deterministic drift. Flowpipe propagation is finally performed through a region of non-Lipschitz continuity.

models, and compare our approach with Flow*, the state-of-the-art verification tool for nonlinear models [32, 33, 35]. We instantiate our approach on top of SpaceEx [60], which is a state-of-the-art tool specialised to linear hybrid models [57, 59, 86]. We evaluate both approaches in safety verification using *flowpipe propagation*, which computes the set of reachable states from a given set of initial states up to a given time horizon. Our experiments demonstrate that our approach performs comparably with Flow* for Lipschitz continuous model, and succeeds with non-Lipschitz models that are out of range for Flow* and violate the working assumptions of many verification tools. These outcomes suggest that neural abstractions are a promising technology, also in view of recent results on direct methods for the safety verification for neural ODEs [66, 67, 93].

We summarise our contributions in the following points:

- we introduce the novel idea of leveraging neural networks to represent abstractions in formal verification, and we instantiate it in safety verification of nonlinear dynamical models;
- we present a CEGIS procedure for the synthesis of neural ODEs that formally overapproximate the dynamics of nonlinear models, which we call neural abstractions;
- we define a translation from neural abstractions defined using ReLU activation functions to hybrid automata with affine dynamics and additive non-determinism;
- we implement our approach[2] and demonstrate its comparable performance w.r.t. the state-of-the-art tool Flow* in safety verification of Lipschitz-continuous models, and even superior efficacy on models that do not exhibit local Lipschitz continuity.

We consider there to be no significant negative societal impact of our work.

---

[2]The code is available at https://github.com/aleccedwards/neural-abstractions-nips22.

## 2 Neural Abstractions of Dynamical Models

We study the formal verification question of whether an $n$-dimensional, continuous-time, autonomous dynamical model with possibly uncertain (bounded) disturbances, considered within a region of interest, is safe with respect to a region of bad states when initialised from a region of initial states.

**Definition 1** (Dynamical Model). *A dynamical model $\mathcal{F}$ defined over a region of interest $\mathcal{X} \subseteq \mathbb{R}^n$ consists of a nonlinear function $f: \mathbb{R}^n \to \mathbb{R}^n$ and a possibly null disturbance radius $\delta \geq 0$. Its dynamics are given by the system of nonlinear ODEs*

$$\dot{x} = f(x) + d, \quad \|d\| \leq \delta, \quad x \in \mathcal{X}, \tag{1}$$

*where $\| \cdot \|$ denotes a norm operator (unless explicitly stated, we assume the norm operator to be given the same semantics across the paper). A trajectory of $\mathcal{F}$ defined over time horizon $T > 0$ is a function $\xi: [0, T] \to \mathbb{R}^n$ that admits derivative at each point in $[0, T]$ such that, for all $t \in [0, T]$, it holds true that $\xi(t) \in \mathcal{X}$ and $\dot{\xi}(t) = f(\xi(t)) + d_t$ for some $\|d_t\| < \delta$. Notably, symbol $d$ in Equation (1) is interpreted as a non-deterministic disturbance that at any time can take any possible value within the bound provided by $\delta$.*

Let the sets $\mathcal{X}_0 \subset \mathcal{X}$ be a region of initial states and $\mathcal{X}_B \subset \mathcal{X}$ be a region of bad states. We say that a trajectory $\xi$ defined over time horizon $T$ is initialised if $\xi(0) \in \mathcal{X}_0$; additionally, we say that it is safe if $\xi(t) \notin \mathcal{X}_B$ for all $t \in [0, T]$; dually, we say that it is unsafe if $\xi(t) \in \mathcal{X}_B$ for some $t \in [0, T]$. The safety verification question for consists of determining whether all initialised trajectories are safe. If this is the case, then we say that the model is safe with respect to $\mathcal{X}_0$ and $\mathcal{X}_B$. If there exist at least one initialised trajectory that is unsafe, then we say that the model is unsafe.

We tackle safety verification by abstraction, that is, we construct an abstract dynamical model that captures all behaviours of the concrete nonlinear model. This implies that if the abstract model is safe then the concrete model is necessarily safe too, and we can thus apply a verification procedure over the abstraction to determine whether the concrete model is safe. Notably, the converse may not hold: lack of safety of the abstract model does not carry over to the concrete model, because our abstraction is an overapproximation. We ultimately obtain a sound (but not complete) safety verification procedure. Our approach synthesises an abstract dynamical model defined in terms a feed-forward neural network with ReLU activation functions and endowed with a bounded non-deterministic disturbance. This can be seen as a neural ODEs [31] augmented with an additive non-deterministic drift that ensures the abstract model to overapproximate the concrete model. To the best of our knowledge, this is the first work to consider neural ODEs with non-deterministic semantics.

Our feed-forward neural network consists of an $n$-dimensional input layer $y_0$, $k$ hidden layers $y_1, \ldots, y_k$ with dimensions $h_1, \ldots, h_k$ respectively, and an $n$-dimensional output layer $y_{k+1}$. Each hidden or output layer with index $i$ are respectively associated matrices of weights $W_i \in \mathbb{R}^{h_i \times h_{i-1}}$ and a vectors of biases $b_i \in \mathbb{R}^{h_i}$. Upon a valuation of the input layer, the value of every subsequent hidden layer is given by the following equation:

$$y_i = \text{ReLU}(W_i y_{i-1} + b_i). \tag{2}$$

Whereas many activation functions exist, we focus our study on ReLU activation functions, applying function $\max\{x, 0\}$ to every element $x \in \mathbb{R}$ of its $h_i$-dimensional argument. Finally, the value of the output layer is given by the affine map $y_{k+1} = W_{k+1} x_k + b_{k+1}$. Altogether, the network results in a function $\mathcal{N}$ whose output is $\mathcal{N}(x) = y_{k+1}$ for every given input $y_0 = x$.

**Definition 2** (Neural Abstraction). *Let $\mathcal{F}$ be a dynamical model given by function $f: \mathbb{R}^n \to \mathbb{R}^n$ and disturbance radius $\delta \geq 0$ and let $\mathcal{X} \subseteq \mathbb{R}^n$ be a region of interest. A feed-forward neural network $\mathcal{N}: \mathbb{R}^n \to \mathbb{R}^n$ defines a neural abstraction of $\mathcal{F}$ with error bound $\epsilon > 0$ over $\mathcal{X}$, if it holds true that*

$$\forall x \in \mathcal{X}: \|f(x) - \mathcal{N}(x)\| \leq \epsilon - \delta. \tag{3}$$

*Then, the neural abstraction consists of the dynamical model $\mathcal{A}$ defined by $\mathcal{N}$ and disturbance $\epsilon$, whose dynamics are given by the following neural ODE with bounded additive disturbances:*

$$\dot{x} = \mathcal{N}(x) + d, \quad \|d\| \leq \epsilon, \quad x \in \mathcal{X}. \tag{4}$$

**Theorem 1** (Soundness of Neural Abstractions). *If $\mathcal{A}$ is a neural abstraction of a dynamical system $\mathcal{F}$ over a region of interest $\mathcal{X} \subseteq \mathbb{R}^n$, then every trajectory of $\mathcal{F}$ is also a trajectory of $\mathcal{A}$.*

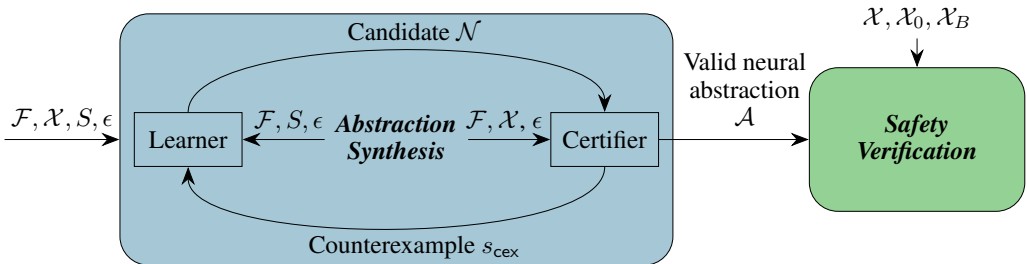

Figure 2: Architecture for the safety verification of nonlinear dynamical models using neural abstractions. The inputs to our architecture are a concrete model $\mathcal{F}$ and its domain of interest $\mathcal{X}$, a finite set of initial datapoints $S$, a desired approximation error $\epsilon$, and regions of initial $\mathcal{X}_0$ and bad states $\mathcal{X}_B$.

*Proof of Theorem 1.* Let $\xi$ be a trajectory of $\mathcal{F}$ and $T$ be the time horizon over which $\xi$ is defined. Then, let $t \in [0, T]$. By definition of trajectory we have that (i) $\xi(t) \in \mathcal{X}$ and there exists $d_t$ s.t. (ii) $\|d_t\| \leq \delta$ and (iii) $\dot{\xi}(t) = f(\xi(t)) + d_t$. By (i) and condition (3) we have that $\|f(\xi(t)) - \mathcal{N}(\xi(t))\| + \delta \leq \epsilon$. Then, by (ii) we have that $\|f(\xi(t)) - \mathcal{N}(\xi(t))\| + \|d_t\| \leq \epsilon$ which, by triangle inequality, implies that $\|f(\xi(t)) + d_t - \mathcal{N}(\xi(t))\| \leq \epsilon$. Using (iii), we rewrite it into $\|\dot{\xi}(t) - \mathcal{N}(\xi(t))\| \leq \epsilon$. Finally, we define $d'_t = \dot{\xi}(t) - \mathcal{N}(\xi(t))$. As a result, we have that $\|d'_t\| \leq \epsilon$ and $\dot{\xi}(t) = \mathcal{N}(\xi(t)) + d'_t$ which, together with (i), shows that $\xi$ is a trajectory of $\mathcal{A}$. $\qquad\square$

**Corollary 1.** *Let $\mathcal{X}_0 \subset \mathcal{X}$ be a region of initial states and $\mathcal{X}_B \subset \mathcal{X}$ and region of bad states. It holds true that if $\mathcal{A}$ is safe with respect to $\mathcal{X}_0$ and $\mathcal{X}_B$ then also $\mathcal{F}$ is safe with respect to $\mathcal{X}_0$ and $\mathcal{X}_B$.*

*Proof of Corollary 1.* By Theorem 1, if there exists an initialised trajectory of $\mathcal{F}$ that is unsafe, then the same is an initialised trajectory of $\mathcal{A}$ that is unsafe. The statement follows by contraposition. $\quad\square$

**Remark 1** (Existence of Neural Abstractions)**.** *Let $\mathcal{F}$ be a dynamical model defined by function $f$ and disturbance radius $\delta \geq 0$, and let $\mathcal{X} \subseteq \mathbb{R}^n$ be a domain of interest. A neural abstraction of $\mathcal{F}$ with arbitrary error bound $\epsilon > 0$ over $\mathcal{X}$ exists if a neural network that approximates $f$ with error bound $\epsilon - \delta$ (cf. condition (3)) exists over the same domain. In this work, we do not prescribe conditions on either width or depth of the network to ensure existence of a neural abstraction. Such conditions are given by universal approximation theorems for neural networks with ReLU activation functions, which have been developed in seminal work in machine learning [23, 40, 61, 72, 88, 91].*

Altogether, we define the neural abstraction of a non-linear dynamical system $\mathcal{F}$ as a neural ODE with an additive disturbance $\mathcal{A}$ that approximates the dynamics while also accounting for the approximation error. Notably, we place no assumptions on the vector field $f$. In particular, Theorem 1 does not require $f$ to be Lipschitz continuous: the soundness of a neural abstraction relies on condition (3), whose certification we offload to an SMT solver (cf. Section 3.2). The resulting neural abstraction is to a hybrid automaton with affine dynamics and non-deterministic disturbance (cf. Section 4), which does not rely on the Picard-Lindelof theorem to ensure uniqueness or existence of a solutions.

## 3 Formal Synthesis of Neural Abstractions

Our approach to abstraction synthesis follows two phases—a *learning* phase and a *certification* phase—that alternate each other in a CEGIS loop [1, 3, 41, 76, 99, 106, 107] (cf. Figure 2, left). Our learning phase trains the parameters of a neural network $\mathcal{N}$ to approximate the system dynamics over a finite set of samples $S \subset \mathcal{X}$ of the domain of interest. Learning uses gradient descent algorithms, which can possibly scale to large amounts of samples. Then, our certification phase either confirms the validity of condition (3) or produces a counterexample which we use to sample additional states and repeat the loop. Certification is based on SMT solving, which reasons symbolically over the continuous domain $\mathcal{X}$ and assures soundness. As a consequence, when certification confirms condition (3) formally valid, then as per Theorem 1 our neural abstraction $\mathcal{A}$ is a sound overapproximation of the concrete model $\mathcal{F}$ and is thus passed to safety verification (cf. Figure 2, right).

Neural networks have been used in the past as representations of formal certificates for the correctness of systems such as Lyapunov neural networks, neural barrier certificates, neural ranking functions

and supermartingales [1, 2, 4, 29, 30, 42, 65, 87, 95, 108, 114–116]. In the present work, we use neural networks for the first time as abstractions, and we instantiate this idea in safety verification of nonlinear models. We shall now present the components of our abstraction synthesis procedure: learner (cf. Section 3.1) and certifier (cf. Section 3.2).

## 3.1 Learning Phase

As with many machine learning-based algorithm, learning neural abstractions hinges on the loss function used as part of the gradient descent scheme for optimising parameters. The task is that of a regression problem, so the choice of loss function to be minimised is simple, namely,

$$\mathcal{L} = \sum_{s \in S} \|f(s) - \mathcal{N}(s)\|_2,$$ (5)

where $\| \cdot \|_2$ represents the $2 - \text{norm}$ of its input, and $S \subset \mathcal{X}$ is a finite set of data points that are sampled from the domain of interest. In other words, the neural abstractions are synthesised using a scheme based on gradient descent to find the parameters that minimise the mean square error over $S$.

The main inputs to the learning procedure are the vector field $f$ of the concrete dynamical model, an initial set of points $S$ sampled uniformly from the domain of interest $\mathcal{X}$. Additional parameters include the hyper-parameters for the learning scheme such as the learning rate, and a stopping criterion for the learning procedure. For the latter, there are two possible options: a target error which all data points must satisfy, or a bound on the value of the loss function. If a target error smaller than $\epsilon - \delta$ is provided, this is when all points in the data set $S$ satisfy the specification (3) and certification subsequently check that this generalises over the entire $\mathcal{X}$. If an alternative loss-based stopping criterion is provided, then an error bound on the approximation is estimated using the maximum approximation error over the data set $S$ for use in certification. This estimated bound is conservative, i.e., greater than the maximum, to allow for successful certification to be more likely.

After learning, the network $\mathcal{N}$ is translated to symbolic form and passed to the certification block, which checks condition (3) as described in Section 3.2. The certifier either determines condition (3) valid, and thus the CEGIS loop terminates, or computes a counterexample that falsifies the condition. The counterexample is returned to the learning procedure and augmented by sampling for additional points nearby in order to maximise the efficiency of learning and the overall synthesis.

## 3.2 Certification Phase

The purpose of the certification is to check that at no point in the domain of interest $\mathcal{X}$ is the maximum error greater than the upper bound $\epsilon - \delta$, as per the specification in condition (3). Therefore, the certifier is provided with the negation of the specification, namely

$$\exists x \colon \underbrace{x \in \mathcal{X} \wedge \|f(x) - \mathcal{N}(x)\| > \epsilon - \delta}_{\phi}.$$ (6)

The certifier seeks an assignment $s_{\mathsf{cex}}$ of the variable $x$ such that the quantifier-free formula $\phi$ is *satisfiable*, namely that the specified bound is violated. If this search is successful, then the network $\mathcal{N}$ has not achieved the specified accuracy over $\mathcal{X}$, and is thus not a valid neural abstraction. The corresponding assignment $s_{\mathsf{cex}}$ forms the counterexample that is provided back to the learner (the machine learning procedure from Section 3.1). Alternatively, if no assignment is found then specification (3) is proven valid; network $\mathcal{N}$ and error bound $\epsilon$ are then passed to the safety verification procedure (cf. Section 4).

Certification of the accuracy of the neural abstractions is performed by an SMT solver. Several options exist for the selection of the SMT solver, with the requirement that the solver should reason over quantifier-free nonlinear real arithmetic formulae [55, 62]. This is because the vector field $f$ may contain nonlinear terms. In our experiments, we employ dReal [62], which supports polynomial and non-polynomial terms such as transcendental functions like trigonometric or exponential ones.

A successful verification process allows for the full abstraction to be constructed using the achieved error $\epsilon$ and neural network $\mathcal{N}$. CEGIS has been shown to perform well and terminate successfully across a wide variety of problems. We demonstrate the robustness of our procedure in Appendix B (see supplementary material).

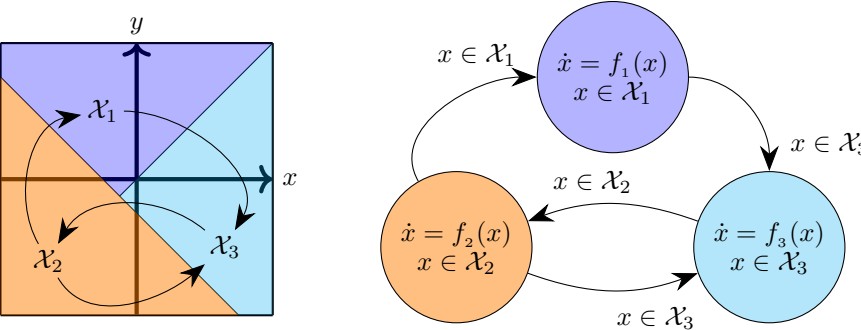

Figure 3: A hybrid automaton corresponding to a state-space partitioning. Each of the three discrete modes corresponds to a unique partition $\mathcal{X}_i$ and vector field $f_i(x)$. Discrete transitions are denoted by the edges of the directed graph with a transition between two modes if the corresponding partitions $\mathcal{X}_i$ and $\mathcal{X}_j$ are adjacent and a trajectory from $f_i$ 'crosses' the corresponding partition.

## 4 Safety Verification of Neural Abstractions

Neural abstractions are dynamical models expressed in terms of neural ODEs with additive disturbances (cf. Equation 4). Corollary 1 ensures the fact for which concluding that a neural abstraction is safe suffices to assert that the concrete dynamical model is also safe. Consequently, once a neural ODE is formally proven to be an abstraction for the concrete dynamical model, which is entirely delegated to our synthesis procedure (cf. Section 3), our definition of neural abstractions enables any procedure for the safety verification of neural ODEs with disturbances to be a valid safety verification procedure for the corresponding dynamical model.

Safety verification approaches for dynamical systems controlled by neural networks solve a similar problem [17,52,73,75,104,110,111,113], yet with a subtle difference: neural network controllers take control actions at discrete points in time. Instead, neural ODEs characterise dynamics over continuous time. Some procedures for the direct verification of neural ODEs have been introduced very recently, and this currently an area under active development [66, 67, 93]. Yet, existing approaches do not consider the case of a neural ODE with a non-deterministic drift. Therefore, in order to obtain a verification procedure for neural abstractions, we build upon the observation that a neural ODEs with ReLU activation functions and non-deterministic drift defines a hybrid automaton with affine dynamics.

Hybrid automata (cf. Figure 3) model the interaction between continuous dynamical systems and finite-state transition systems [69, 112]. A hybrid automaton consists of a finite set of variables and a finite graph, whose vertices we call discrete modes and edges we call discrete transitions. Every mode is associated with an invariant condition and a flow condition over the variables, which determine the continuous dynamics of the systems on the specific mode. Every discrete transition is associated with a guard condition, which determines the effect on discrete transitions between modes. While we refer the reader to seminal work for a general definition of hybrid automata [69], we present a translation from neural abstractions to hybrid automata.

### 4.1 Translation of Neural Abstractions Into Hybrid Automata

We begin with the observation that each neuron within a given hidden layer of a neural network with ReLU activation functions induces a hyperplane in the vector space associated with the previous layer This hyperplane results in two half-spaces, one corresponding to the neuron being active and one to it being inactive. For the $j$th neuron in the $i$th layer, these two halfspaces are respectively the two parts of the hyperplane given by

$$\{y_{i-1} \mid W_{i,j}y_{i-1} + b_{i,j} = 0\}, \tag{7}$$

where $W_{i,j}$ is the $j$th row of the weight matrix $W_i$ and $b_{i,j}$ is the $j$th element of the bias $b_i$ (cf. Section 2). Therefore, every combinatorial configuration of the neural network defines an intersection of halfspaces that defines a polyhedral region in the vector space of the input neurons. Moreover, every

configuration also defines a linear function from input to output neurons. The space of configurations thus defines a partitioning of the input space, where each region is associated with an affine function. A neural abstraction casts into a hybrid automaton, where every mode is determined by a configuration of the hidden neurons and each of these configurations induces a system of affine ODEs (cf. Figure 3).

**Discrete Modes**  We represent a configuration of a neural network as a sequence $C = (c_1, \ldots, c_k)$ of Boolean vectors $c_1 \in \{0,1\}^{h_1}, \ldots, c_k \in \{0,1\}^{h_k}$, where $k$ denotes the number of hidden layers and $h_1, \ldots, h_k$ denote the number neurons in each of them (cf. Section 2). Every vector $c_i$ represents the configuration of the neurons at the $i$th hidden later, and the $j$th element of $c_i$ represent the activation status of the $j$th neuron at the $i$th later, which equals to 1 is the neuron is active and 0 if it is inactive. Every mode of the hybrid automaton corresponds to exactly one configuration of neurons.

**Invariant Conditions**  We define the invariant of each mode as a restriction of the domain of interest to a region $\mathcal{X}_C \subseteq \mathcal{X}$, which denotes the maximal set of states that enables configuration $C$. To construct $\mathcal{X}_C$, we define a higher-dimensional polyhedron on the space of valuation of the neurons that enable configuration $C$, i.e.,

$$\mathcal{Y}_C = \left\{ (y_0, \ldots, y_k) \ \middle| \ \begin{array}{l} \wedge_{i=1}^k y_i = \mathrm{diag}(c_i)(W_i y_{i-1} + b_i) \wedge \\ \mathrm{diag}(2c_i - 1)(W_i y_{i-1} + b_i) \geq 0 \end{array} \right\}. \tag{8}$$

Note that $\mathrm{diag}(v)$ denotes the square diagonal matrix whose diagonal takes its coefficients from vector $v$; in our case, this results in a square diagonal matrix whose coefficients are either 0 or 1. Then, we project $\mathcal{Y}_C$ onto the input neurons $y_0$, denoted $\mathcal{Y}_C \restriction_{y_0}$. Since the input neurons $y_0$ are equivalent to the state variables of the dynamical model, the invariant condition of mode $C$ results in

$$\mathcal{X}_C = (\mathcal{Y}_C \restriction_{y_0}) \cap \mathcal{X}. \tag{9}$$

A projection can be computed using the Fourier-Motzkin algorithm or by projecting the vertices of the polyhedron in a double description method. However, even though this is effective in our experiments, it has worst-case exponential time complexity. A polynomial time construction can be obtained by propagating halfspaces backwards along the network, similarly to methods used in abstraction-refinement [27, 58]. We outline the alternative construction in Appendix C.1.

**Flow Conditions**  The dynamics of each mode $C$ can be seen itself as a dynamical system with bounded disturbance:
$$\dot{x} = A_C x + b_C + d, \quad \|d\| \leq \epsilon, \quad x \in \mathcal{X}_C. \tag{10}$$
The matrix $A_C \in \mathbb{R}^{n \times n}$ and the vector of drifts $b_C \in \mathbb{R}^n$ determine the linear ODE of the mode, whereas $\epsilon > 0$ is the error bound derived from the neural abstraction. The coefficients of the system are given by the weights and biases of the neural network as follows:

$$A_C = W_{k+1} \prod_{i=1}^k \mathrm{diag}(c_i) W_i, \tag{11}$$

$$b_C = b_{k+1} + \sum_{i=1}^k (W_{k+1} \prod_{j=i+1}^k \mathrm{diag}(c_j) W_j) \mathrm{diag}(c_i) b_i. \tag{12}$$

**Discrete Transitions and Guard Conditions**  A discrete transition exists between any two given modes if the two polyhedra that define their invariant conditions share a facet and the dynamics pass through at some point along the facet. This can be checked by considering the sign of the Lie derivative between the dynamics and the corresponding facet, that is, the inner product between the dynamics and the normal vector to the facet. In practice, we take a faster but more conservative approach by considering that a transition exists between two modes when the corresponding polyhedral regions share at least a vertex. The guard condition of a discrete transition is simply the invariant of the destination mode.

## 4.2 Enumeration of Feasible Modes

A given configuration $C$ exists in the hybrid automaton if and only if the corresponding set $\mathcal{X}_C$, which is a convex polyhedron in $\mathbb{R}^n$, is nonempty; this consists of verifying that the linear program (LP) constructed from the polyhedron is feasible. Finding all modes of the hybrid automaton therefore consists of solving $2^H$ linear programs, where $H = h_1 + \cdots + h_k$ is the total number of hidden neurons in the network. However, this exponential scaling with the number of neurons is limiting

in terms of network size. Therefore, we propose an approach that works very well in practice to determine all valid neuron configurations.

The approach relies on the observation that within a bounded polyhedron $\mathcal{P}$, a given neuron has two modes (ReLU enabled or disabled) only if the induced hyperplane intersects $\mathcal{P}$. If it does not, only one of the two possible half-spaces contributes to any possible active configuration, and the other neuron mode can be disregarded. Therefore, this approach involves iterating through each neuron in turn and constructing two LPs—one for each halfspace intersected with the domain of interest $\mathcal{X}$. If only one LP is valid, we can fix the neuron to this mode, i.e., from this point onward only consider the intersection with the halfspace corresponding to the feasible LP, and construct a new polyhedron from the intersection of $\mathcal{X}$ and the feasible half-space. In short, we consider the neurons of the network as a binary tree, with the branches representing the enabled and disabled state of this neuron. We perform a depth-first tree search through this tree by intersecting with the corresponding half-spaces. Upon reaching an end node, we store this configuration (branches taken) and revert back to the most recent unexplored branch and continue. We include a more detailed description of this algorithm in Appendix C.2. This approach is inspired by that presented in [21], which similarly enumerates through the path of neurons using sets to determine the output range of a network.

## 5 Experimental Results

### 5.1 Safety Verification Using Neural Abstractions

We benchmark the results obtained by the safety verification algorithm proposed in Section 4 against Flow* [33] (available under GPL), which is a mature tool for computing reachable regions of hybrid automata. It relies on computing flowpipes, i.e., sets of reachable states across time, which are propagated from a given set of initial states. The flowpipes are generated from Taylor series approximations of the model's vector field in (1), over subsequent discrete time steps. Crucially, the use of a higher-order Taylor series, or of smaller time steps, leads to more precise computation of reachable sets. Since Flow*, like SpaceEx (available under GPLv3) is able to calculate over-approximations of flowpipes, it is suitable for use in safety verification, and is a state-of-the-art tool for verifying safety of nonlinear models.

Making a fair comparison around metrics for accuracy between Flow* and SpaceEx is challenging, as they represent flowpipes differently [36]. We ask them to perform safety verification for a given pair of initial and bad states, on a collection of different nonlinear models. These models, and their parameters, are detailed in Appendix A. As described in Section 2, the task of safety verification consists of ensuring that no trajectory starting within the set of initial states enters the set of bad states, within a given time horizon.

Our setup is as follows. Firstly, for a given benchmark model we define a finite time horizon $T$, a region of initial states $\mathcal{X}_0$ and a region of bad states $\mathcal{X}_B$. Then, we run flowpipe computations with Flow* using high-order Taylor models. Similarly we run the procedure described in Section 3, and construct a hybrid automaton as described in Section 4 to perform flowpipe computations using SpaceEx. We present the results in Table 1. In the table, we show the Taylor model order (TM) and time step used within Flow*, as well as the structure of the neural networks used for neural abstractions. For example, we denote a network with two hidden layers with $h_1$ neurons in the first layer and $h_2$ neurons in the second hidden layer as $[h_1, h_2]$. We note that while Flow*, much like SpaceEx, can perform flowpipe computation on the constructed hybrid automaton, it is not specialised to linear models like SpaceEx is and in practice struggles with the number of modes.

Notably, Flow* is unable to handle the two models that do not exhibit local Lipschitz continuity. Flow* constructs Taylor models that incorporate the derivatives of the dynamics: as expected, unbounded derivatives will cause issues for this approach. Meanwhile, Ariadne [22] a is an alternative tool for over-approximating flowpipes of nonlinear systems. While Ariadne does not explicitly require Lipschitz continuity, it is also unable to perform analysis on tools with $nth$ root terms at zero, due to numerical instability. Instead, our abstraction method works directly on the dynamics themselves, rather than their derivatives, in order to construct simpler, abstract models that are amenable to be verified. By formally quantifying how different an abstract model is through the approximation error, we are able to formally perform safety verification on such challenging concrete models.

Table 1: Comparison of safety verification between Flow* and the combination of Neural Abstractions plus SpaceEx. Here, $T$: time horizon, $TM$: Taylor model order, $\delta$: time-step, $t$: total computation time (better times denoted by **bold**), $W$: network neural structure, $M$: total number of modes in resulting hybrid automaton, $Blw$: blowup in the error before $T$ is reached, and -: no results unobtainable.

| Model | $T$ | Flow* | | | | Neural Abstractions | | | |
| --- | --- | --- | --- | --- | --- | --- | --- | --- | --- |
| | | TM | $\delta$ | Safety Ver. | $t$ | $W$ | $M$ | Safety Ver. | $t$ |
| Jet Engine | 1.5 | 10 | 0.1 | Yes | **1.3** | [10, 16] | 8 | Yes | 215 |
| Steam Governor | 2.0 | 10 | 0.1 | Yes | **62** | [12] | 29 | Yes | 219 |
| Exponential | 1.0 | 30 | 0.05 | Blw | 1034 | [14, 14] | 12 | Yes | **308** |
| Water Tank | 2.0 | - | - | No | - | [12] | 6 | Yes | **49** |
| Non-Lipschitz 1 | 1.4 | - | - | No | - | [10] | 12 | Yes | **19** |
| Non-Lipschitz 2 | 1.5 | - | - | No | - | [12, 10] | 32 | Yes | **59** |

Notice that we additionally outperform Flow* on a Lipschitz-continuous model (*Exponential* in Table 1), where the composition of functions that make up the model's dynamics result in high errors in Flow* before the flowpipe can be calculated across the given time horizon. We highlight that despite relying on affine approximations (i.e., 1st order models), neural abstractions are able to compete with, and even outperform, methods that use much higher order functions (10th and 30th in the benchmarks) for approximation.

## 5.2 Limitations

Our approach is limited in terms of scalability, both with regards to the dimension of the models and to the size of the utilised neural networks. The causes of this limitation are twofold: firstly we are bound by the computational complexity of SMT solving - known to be NP-hard - which can struggle with complex formaulae with many variables. The certification step requires the largest amount of time (cf. Appendix B), indicating that improvements in the verification of neural networks can lead to a large performance increase for our abstractions.

Secondly, we are limited in terms of the complexity of our abstractions by SpaceEx. While SpaceEx is a highly efficient implementation of LGG [86], the presence of a large number of discrete modes poses a significant computational challenge. It future work, we hope to investigate the balance between abstraction complexity and accuracy. The efficacy of neural abstraction on further tools for hybrid automata with affine dynamics also remains to be investigated [6, 22, 26, 105].

## 6 Conclusion

We have proposed a novel technique that leverages the approximation capabilities of neural networks with ReLU activation functions to synthesise formal abstractions of dynamical models. By combining machine learning and SMT solving algorithms in a CEGIS loop, our method computes abstract neural ODEs with non-determinism that overapproximate the concrete nonlinear models. This guarantees the property for which safety of the abstract model carries over to the concrete model. Our method casts these neural ODEs into hybrid automata with affine dynamics, which we have verified using SpaceEx. We have demonstrated that our method is not only comparable to Flow* in safety verification on existing nonlinear benchmarks, but also shows superior effectiveness on models that do not exhibit local Lipschitz continuity, which is a hard problem in formal verification. Yet, our experiments are limited to low-dimensional models and scalability remains an open challenge. Our approach has advanced the state of the art in terms of expressivity, which is the first step toward obtaining a general and efficient verifier based on neural abstraction. Obtaining scalability to higher dimensions will require a synergy of efficient SMT solvers for neural networks and safety verification of neural ODEs, which are both novel and actively researched questions in formal verification [66,67,74,77,90,93,111].

## Acknowledgements

We thank the anonymous reviewers for their helpful suggestions. Alec was supported by the EPSRC Centre for Doctoral Training in Autonomous Intelligent Machines and Systems (EP/S024050/1).

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
