# Neural Abstractions: Supplementary Material

**Alessandro Abate**[*]
Department of Computer Science
University of Oxford, UK

**Alec Edwards**[*]
Department of Computer Science
University of Oxford, UK

**Mirco Giacobbe**[*]
School of Computer Science
University of Birmingham, UK

## A  Benchmark Nonlinear Dynamical Models

For each dynamical model, we report the vector field $f : \mathbb{R}^n \to \mathbb{R}^n$ and the spatial domain $\mathcal{X}$ over which the abstraction is performed and which, unless otherwise stated, is taken to be the hyper-rectangle $[-1,1]^n$.

**Water Tank**

$$\begin{cases} \dot{x} = 1.5 - \sqrt{x} \\ \mathcal{X}_0 = [0, 0.01] \\ \mathcal{X}_B = \{x | x \geq 2\} \end{cases} \tag{1}$$

**Jet Engine** [2]

$$\begin{cases} \dot{x} = -y - 1.5x^2 - 0.5x^3 - 0.1, \\ \dot{y} = 3x - y, \\ \mathcal{X}_0 = [0.45, 0.50] \times [-0.60, -0.55] \\ \mathcal{X}_B = [0.3, 0.35] \times [0.5, 0.6] \end{cases} \tag{2}$$

**Steam Governor** [3]

$$\begin{cases} \dot{x} = y, \\ \dot{y} = z^2 \sin(x) \cos(x) - \sin(x) - 3y, \\ \dot{z} = -(\cos(x) - 1), \\ \mathcal{X}_0 = [0.70, 0.75] \times [-0.05, 0.05] \times [0.70, 0.75] \\ \mathcal{X}_B = [0.5, 0.6] \times [-0.4, -0.3] \times [0.7, 0.8] \end{cases} \tag{3}$$

**Exponential**

$$\begin{cases} \dot{x} = -\sin(\exp(y^3 + 1)) - y^2 \\ \dot{y} = -x, \\ \mathcal{X}_0 = [0.45, 0.5] \times [0.86, 0.91] \\ \mathcal{X}_B = [0.3, 0.4] \times [0.5, 0.6] \end{cases} \tag{4}$$

---

[*]The authors are listed alphabetically

36th Conference on Neural Information Processing Systems (NeurIPS 2022).

**Non-Lipschitz Vector Field 1 (NL1)**

$$\begin{cases} \dot{x} = y \\ \dot{y} = \sqrt{x} \\ \mathcal{X} = [0,1] \times [-1,1], \\ \mathcal{X}_0 = [0,0.05] \times [0,0.1] \\ \mathcal{X}_B = [0.35,0.45] \times [0.1,0.2] \end{cases} \tag{5}$$

**Non-Lipschitz Vector Field 2 (NL2)**

$$\begin{cases} \dot{x} = x^2 + y \\ \dot{y} = \sqrt[3]{x^2} - x, \\ \mathcal{X}_0 = [-0.025,0.025] \times [-0.9,-0.85] \\ \mathcal{X}_B = [-0.05,0.05] \times [-0.8,-0.7] \end{cases} \tag{6}$$

# B   Additional Experimental Results and Figures

## B.1   Experimental Comparison Against Affine Simplical Meshes

In this section, we present some supplementary empirical results on neural abstractions. Firstly, we note that hybridisation-based abstraction of nonlinear models have been studied previously, such as in [1], which describes a type of hybridisation-based abstractions that is similar to those constructed in this work. The approach relies first on partitioning the state space using a simplicial mesh grid, and then allowing the dynamics in each mesh to be calculated from an affine interpolation between the vertices of the simplex. This affine simplical mesh (ASM) based approach constructs abstractions of the same expressivity as neural abstractions (first order approximations) with partitions defined by affine inequalities. An approximation-error bound for ASM can be calculated for systems which have bounded second order derivatives using the model dynamics and the size of each simplex (all simplices are assumed to be the same size), as described in [1]. In Table 2 we compare between abstractions constructed using an affine simplical mesh and neural abstractions. We run our procedure to synthesise certified abstractions using selected network structures and an initial target error of 0.5. If a successful abstraction is synthesised, we reduce the error by some multiplicative factor and repeat. This iterative procedure continues until no success is reached within a time of 300s. We report the results from 10 repeated experiments over different initial random seeds for neural abstractions, reporting the average (mean), minimum and maximum results obtained. In contrast, we report the approximation-error bound for ASM for different numbers of partitions.

The results reported in Table 2 illustrate that neural abstractions outperform ASM based abstractions in terms of error for similar numbers of partitions. Furthermore, neural abstractions generally require significantly fewer partitions for significantly lower approximation-error bounds. In practice this means neural abstractions will outperform ASM-based abstractions for safety verification both in terms of speed and accuracy. We also note the success ratio of our experiments, i.e., the ratio of all experiments which achieve an approximation-error bound of 0.5 or less. These results suggest that in general or procedure is robust and terminates successfully with high probability for reasonable target errors.

We note that since ASM based abstractions are constructive and are able to deterministically increase the number partitions and consequently reduce the error, for very large numbers of partitions they would achieve lower errors than neural abstractions. However, in practice these abstractions would be too large in complexity to use with SpaceEx for safety verification.

## B.2   Computation Run-time Profiling

In Table 3 we show a breakdown of the runtimes of our procedure shown in the main text. In particular, we present the total time spent during learning, certification of the abstraction and finally in safety verification.

Table 2: A comparison between abstractions constructed using an affine simplicial mesh and neural abstractions. Here, $W$ represents the neural structure used for neural abstraction, $N_P$: total number of partitions, $\epsilon$: the calculated upper bound on the approximation error, $\bar{N}_P$: average (mean) number of partitions, $\bar{\epsilon}$: average (mean) approximation error bound, $\epsilon^+$ : the maximum approximation error, $\epsilon^-$: the minimum approximation error, Success Ratio: the ratio of repeated experiments that terminated successfully (i.e., an error of 0.5 was reached within the first timeout of 300s). Note, we only include successful experiments when calculating the average, min and max (since no error exists for unsuccessful experiments). All reported errors use the `2-norm`.

| Benchmark | Affine Simplicial Mesh | | Neural Abstractions | | | | | |
| | $N_p$ | $\epsilon$ | $W$ | $\bar{N}_P$ | $\bar{\epsilon}$ | $\epsilon^+$ | $\epsilon^-$ | Success Ratio |
| --- | --- | --- | --- | --- | --- | --- | --- | --- |
| Jet Engine | 8 | 1.33 | [10] | 9 | 0.11 | 0.22 | 0.040 | 1.0 |
| | 32 | 0.33 | [10, 10] | 27 | 0.077 | 0.17 | 0.040 | 1.0 |
| | 128 | 0.083 | [15, 15] | 61 | 0.058 | 0.071 | 0.053 | 1.0 |
| Steam | 24 | 3.58 | [10] | 27 | 0.27 | 0.37 | 0.21 | 1.0 |
| | 192 | 0.89 | [20] | 236 | 0.18 | 0.27 | 0.15 | 1.0 |
| Exponential | 8 | 13.7 | [10] | 9 | 0.29 | 0.40 | 0.22 | 0.5 |
| | 32 | 3.44 | [20] | 30 | 0.19 | 0.22 | 0.13 | 0.9 |
| | 128 | 0.86 | [20, 20] | 75 | 0.15 | 0.22 | 0.071 | 1.0 |

Table 3: Breakdown of the timings shown in Table 1. Shown are the timings in the constituent component shown in Figure 2: time spent during learning, time spent during certification of the neural abstraction, and time spent during safety verification. Remaining time is spent in overheads, such as converting from neural network to hybrid automaton.

| Model | Learner | Certifier | Safety Verification |
| --- | --- | --- | --- |
| Jet Engine | 19 | 194 | 1.8 |
| Steam Governor | 42 | 177 | 0.5 |
| Exponential | 27 | 278 | 3.3 |
| Water-tank | 48 | 0.001 | 0.05 |
| Non-Lipschitz 1 | 13 | 0.50 | 5.5 |
| Non-Lipschitz 2 | 31 | 15 | 5.1 |

# C Improved Translation from Neural Abstractions to Hybrid Automata

## C.1 Computing Invariant Conditions

Invariant conditions are computed from the configuration of a neural network denoted as the sequence $C = (c_1, \ldots, c_k)$ of Boolean vectors $c_1 \in \{0, 1\}^{h_1}, \ldots, c_k \in \{0, 1\}^{h_k}$, where $k$ denotes the number of hidden layers and $h_1, \ldots, h_k$ denote the number neurons in each of them (cf. Section 2). Every vector $c_i$ represents the configuration of the neurons at the $i$th hidden later, and its $j$th element $c_{i,j}$ represents the activation status of the $j$th neuron at the $i$th layer. Every mode of the hybrid automaton corresponds to exactly one configuration of neurons. In turn, every configuration of neurons $C$ restricts the neural network $\mathcal{N}$ into a linear function. More precisely, we inductively define the linear restriction at the $i$th hidden layer as follows:

$$\mathcal{N}_C^{(i)}(x) = \text{diag}(c_i)(W_i \mathcal{N}_C^{(i-1)}(x) + b_i), \text{ for } i = 1, \ldots, k, \qquad \mathcal{N}_C^{(0)}(x) = x. \qquad (7)$$

We define the invariant of each mode as a restriction of the domain of interest to a region $\mathcal{X}_C \subseteq \mathcal{X}$, which denotes the maximal set of states that enables configuration $C$. To construct $\mathcal{X}_C$, we begin with the observation that the activation configuration $c_i$ at every $i$th hidden layer induces a halfspace on the vector space of the previous layer of the neural network. Then, the pre-image of this halfspace backward along the previous layers of the linear restriction of the network characterises a corresponding halfspace on its input neurons. Since the input neurons are equivalent to the state

variables of the dynamical model, the halfspace induced by layer $i$ projected onto state variables $x$ is

$$\mathcal{H}_C^{(i)} = \text{pre-image of } \underbrace{\{y_{i-1} \mid \text{diag}(2c_i - 1)(W_i y_{i-1} + b_i) \geq 0\}}_{\text{halfspace induced by } i\text{th layer onto } (i-1)\text{th layer}} \text{ under } \mathcal{N}_C^{(i-1)} \qquad (8)$$

The pre-image of a set $\mathcal{Y}$ under a function $g$ is defined as $\{x \mid g(x) \in \mathcal{Y}\}$ and can be generally computed by quantifier elimination or, in the linear case, double description methods. However, these methods have worst-case exponential time complexity. To obtain $\mathcal{X}_C$ efficiently, we can leverage the fact that the pre-image of any halfspace $\{y \mid c^{\mathsf{T}} y \leq d\}$ under any affine function $g(x) = Ax + b$ equals to the set $\{x \mid c^{\mathsf{T}} y \leq d \wedge y = Ax + b\}$, which in turn defines the halfspace $\{x \mid c^{\mathsf{T}} Ax \leq d - c^{\mathsf{T}} b\}$. Therefore, since $\mathcal{N}_C^{(i-1)}$ is an affine function, every halfspace can be projected backward through the affine functions $\mathcal{N}_C^{(i-1)}, \ldots, \mathcal{N}_C^{(1)}$ using $\mathcal{O}(k)$ linear algebra operations. Finally, the entire invariant condition for configuration $C$ is defined as the following polyhedron:

$$\mathcal{X}_C = \cap \{\mathcal{H}_C^{(i)} \mid i = 1, \ldots, k\} \cap \mathcal{X}. \qquad (9)$$

An invariant condition thus results in a polyhedron defined as the intersection of $k$ halfspaces together with the constrains that define the domain of interest. Notably, under the definition in this appendix, the dynamics of mode $C$ given in Equation 10 correspond to the affine dynamical model

$$\dot{x} = \mathcal{N}_C^{(k+1)}(x) + d, \quad \|d\| \leq \epsilon, \quad x \in \mathcal{X}_C, \qquad (10)$$

whose dynamics are governed by the affine function

$$\mathcal{N}_C^{(k+1)}(x) = W_{k+1} \mathcal{N}_C^{(k)}(x) + b_{k+1}. \qquad (11)$$

## C.2 Enumerating Feasible Modes

Determining whether a mode $C$ exists in the hybrid automaton amounts to determining the linear program (LP) associated to polyhedron $\mathcal{X}_C$ is feasible. Finding all modes therefore consists of solving $2^H$ linear programs, where $H = h_1 + \cdots + h_k$ is the total number of neurons. This scales exponentially in the number of neurons. Here, we elaborate on the tree search algorithm described in Section 4.2 using a diagram; the purpose of this algorithm is to efficiently determine all active neuron configurations within a bounded domain of interest $\mathcal{X}$.

We consider an example tree in Figure C.1, which depicts an example search for a neural network with a single hidden layer consisting of three neurons. The tree illustrates the construction of $\mathcal{X}_C$ through repeated intersections of half-spaces as paths are taken through the tree structure. Nodes represent each neuron, labelled $N_i$, $i = 1, 2, 3$ and each edge represents one of two possible half-spaces for the neuron it leaves from (ReLU enabled, solid line, and disabled, dashed line). This approach allows us to prune neurons and overall solve significantly fewer linear programs than simply enumerating through all possible configurations.

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

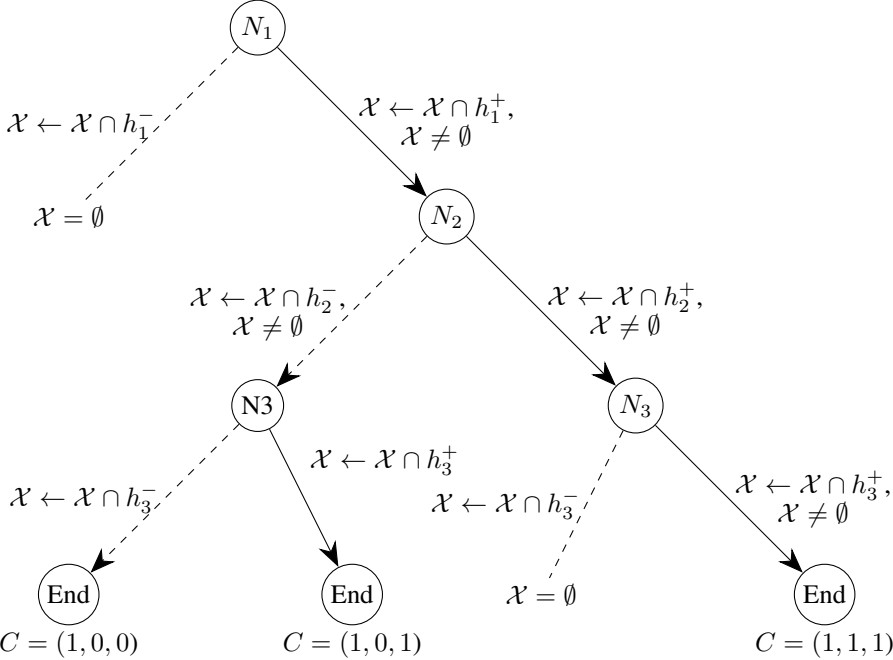

Figure C.1: Example Tree search to determine the active configurations for a neural network consisting of a single hidden layer with 3 neurons. Here, $h_i^+$ denotes the positive half-space ($\{x : w_i x + b_i \geq 0\}$) and $h_i^-$ denotes the negative half-space ($\{x : w_i x + b_i \leq 0\}$) of the $i^{\text{th}}$ neuron; $w_i$ represents the $i^{\text{th}}$ row of the weight matrix corresponding to the hidden layer, and $b_i$ represents the $i^{\text{th}}$ element of the bias vector of the hidden layer. Notably, when the set $\mathcal{X}$ becomes empty, it is no longer necessary to continue along that path. Once we reach the end of the tree, we have an active configuration $C$, and backtrack to the last node that was not fully explored.