# OpenReview forum: "Neural Abstractions"
_NeurIPS.cc/2022/Conference — NeurIPS 2022 Accept_

### Official Review · Reviewer_93rE · 2022-07-11

**Rating:** 6
**Confidence:** 4
**Soundness:** 3 good
**Presentation:** 3 good
**Contribution:** 3 good

**Summary:**

The paper proposes a novel method of using neural-network-based abstractions to verify non-linear dynamical systems. The proposed method can generalize to systems that do not exhibit local Lipschitz continuity.

**Questions:**

- Why not use Flow* or other hybrid automatons analysis methods to analyze the cast hybrid automatons?
- How expensive it is to use dReal in the loop of training?

**Limitations:**

The author provides a detailed discussion of limitations.

**Strengths And Weaknesses:**

The paper proposes a method for learning neural abstractions of non-linear dynamical systems. The work is generally sound and interesting.

## Strengths
- The problem of safety verification on non-linear dynamical systems is a problem of great interest to the community of control. The authors clearly describe the limitations of previous methods and provide a thorough comparison in the experiments.
- The technical parts of this paper are quite strong. It is novel to leverage neural networks to approximate dynamics in safety verification.

However, I have some concerns about the proposed method. I summarize all of my major concerns below:
## Weakness
- Some analysis on the size of hybrid automatons cast from neural abstractions is missing. The size of the hybrid automatons is critical for safety verification since it is directly related to approximation precisions.
- It is not very clear why SpaceEx is used for analyzing the generated hybrid automatons. To the best of my knowledge, Flow* can also handle hybrid automatons.
- Though the authors discussed scalability issues of the proposed method, more detailed experimental results on scalability analysis might be a good add-on. In addition, profiling time costs for different steps in the proposed method is important to support the claims in the paper.

---

> ### Author Response · Authors · 2022-08-02
> **Response to 93rE**
>
> Flow* can indeed be used to verify the cast hybrid automata. We arbitrarily chose SpaceEx because it is specialized (and limited) to hybrid systems with affine ODEs. This choice was made to stress the fact that neural abstractions allow us to verify nonlinear and even non-Lipchitz models using specific tools for piecewise linear systems. The main focus of this paper is on the expressivity of neural abstractions, and secondarily on their performance. A performance comparison between Flow* and SpaceEx on hybrid automata with linear ODEs would be interesting but collateral to our argument.
>
> The SMT-based certification part is in general NP-hard. In this context, its performance depends on the size of the network and the target error bound (a smaller error bound is harder to prove). We have updated the supplementary material to include a breakdown of these time costs for each stage (synthesis - training and certification- and safety verification), in addition to including the size of the automaton in terms of the total number of modes in the main manuscript. The table illustrates that the majority of time is spent during the certification phase. This indicates that further improvements in the verification of neural nets (an active area of research) in combination with nonlinear systems will lead to a large performance increase for our approach.

---

> > ### Comment · Reviewer_9jBF · 2022-08-04
> > **.**
> >
> > "A performance comparison between Flow* and SpaceEx on hybrid automata with linear ODEs would be interesting but collateral to our argument."
> >
> > While I tend to agree with this sentiment, I also totally understand where 93rE comes from. Why bother with this new approach if it doesn't work 'better'.

---

> > > ### Author Response · Authors · 2022-08-04
> > > **Response to 9jBF**
> > >
> > > Our paper has shown that neural abstractions work better than the state of the art on models that do not exhibit local Lipschitz continuity. We remark that models of this kind are common in scientific literature, though are rarely treated in verification (see reply to Cxvo).
> > >
> > > Ultimately, our method—neural abstraction—translates a nonlinear ODE into a hybrid automaton with linear dynamics through abstraction. We demonstrated that verifying over the abstraction, rather than verifying the nonlinear ODE directly, enables an effective verification of non-Lipschitz models. Our contribution is the definition of this abstraction, and the workflow to construct it (CEGIS and casting to hybrid automaton) and exploit it (hybrid automata verification). However, we do not advocate for the use of specific tools as underlying technologies. Comparison between verification tools for hybrid automata with linear dynamics is well treated as part of an annual competition [5]. We chose to use SpaceEx for this purpose, as opposed to CORA [6], Hydra [7], Juliareach [8] and Flow*. Similarly, we chose PyTorch for training as opposed to TensorFlow, Keras or Jax, and we chose dReal for certification as opposed to iSAT [9].
> > >
> > > [5] Althoff, Matthias, et al. "ARCH-COMP21 Category Report: Continuous and Hybrid Systems with Linear Continuous Dynamics." ARCH@ADHS. 2021.
> > >
> > > [6] M. Althoff, ‘An introduction to CORA 2015’, in ARCH@ADHS, 2015
> > >
> > > [7] S. Schupp, et al., ‘HyPro: A C++ Library of State Set Representations for Hybrid Systems Reachability Analysis’, in NFM, 2017
> > >
> > > [8] S. Bogomolov, M. Forets, G. Frehse, K. Potomkin, and C. Schilling, ‘JuliaReach: a toolbox for set-based reachability’, in HSCC, 2019.
> > >
> > > [9] M. Fränzle, C. Herde, T. Teige, S. Ratschan, and T. Schubert, ‘Efficient solving of large non-linear arithmetic constraint systems with complex boolean structure’, in Journal on Satisfiability, Boolean Modeling and Computation, 2007.

---

> > ### Comment · Reviewer_93rE · 2022-08-05
> > **Thanks for the response and additional results**
> >
> > I appreciate authors' response and additional results on time costs.
> >
> > - "arbitrarily chose SpaceEx"
> >
> > Though investigating the performance difference between Flow* and SpaceEx is not the focus of the paper, the paper directly compare Neural Abstraction + SpaceEx with Flow*. I feel it would be more natural to exclude other possible factors when showing the benefits of neural abstractions. Specifically, having the same setting for neural abstraction + Flow* and Flow* would eliminate any benefits introduced from SpaceEx and strengthen the experimental results.
> >
> > - "Breakdown time"
> >
> > Does the learning time include the time for finding counter examples?
> >
> > - "Scalability"
> >
> > Can you comment on how neural abstractions can scale to systems with high-dimensional states?

---

> > > ### Author Response · Authors · 2022-08-08
> > > **Response to 93rE**
> > >
> > > Q1. Our choice of using SpaceEx justified by the fact that SpaceEx is the state of the art in verification of hybrid automata with linear dynamics; conversely, Flow* is the state of the art in verification of nonlinear systems. Nonlinear systems subsume linear systems, therefore Flow* does in principle support hybrid automata with linear dynamics. However, SpaceEx is usually much faster than Flow* on hybrid systems with linear dynamics. On reviewers' advice, we performed the additional comparison between Flow* and SpaceEx on the hybrid automata produced by neural abstraction. The experiment shows that Flow* cannot handle the large number of modes required for neural abstractions.
> > >
> > > SpaceEx owes its performance to efficient methods that exclusively apply to the linear case [12]; notably, SpaceEx handles the higher number of modes better than Flow*, but it is specialised and limited to linear systems. The experiment demonstrates that Neural Abstraction + SpaceEx (specialised and efficient method for linear system) is effective, as opposed to Neural Abstraction + Flow* (general method for nonlinear systems). By translating nonlinear ODEs to hybrid automata with linear ODEs, neural abstraction allows us to gain the advantage introduced by SpaceEx, which is the purpose of our work and it would not be possible otherwise.
> > >
> > > Q2. The time for finding counterexamples corresponds to the certification phase.
> > >
> > > Q3. Scalability to high-dimensional states for nonlinear dynamical systems is an open problem in formal verification. In the annual competition for this category [11] the largest system has 7 variables. While dimensionality plays an important role, several other factors determine hardness. Notably, the hardest problem in the competition is a system with 3 variables. That being said, we remark in section 5.2 that the scalability of our workflow w.r.t. the number of variables in the ODE is bound by the scalability of the certification phase (the SMT solver dReal). Obtaining scalable SMT procedures for neural networks with nonlinear theories is an open and important problem in formal methods. Neural abstraction contributes to the relevance of this technical underlying problem.
> > >
> > > Our message is that neural networks—thanks to their expressive power—enable our method to improve the verification state of the art in terms of expressivity. This even enables the safety analysis of non-Lipschitz systems, which is a very hard problem in verification. We have introduced a workflow and shown this expressivity using existing technology. This is only the first step toward obtaining a general and efficient verifier based on neural abstraction. Obtaining scalability to high-dimensions will require a synergy of progress in SMT for neural networks and verification of neural ODEs, which are very novel problems in formal verification.
> > >
> > >
> > > [11] Geretti, Luca, et al. "ARCH-COMP21 Category Report: Continuous and Hybrid Systems with Nonlinear Dynamics." ARCH@ ADHS. 2021.
> > >
> > > [12] Guernic, Colas Le, and Antoine Girard. "Reachability analysis of hybrid systems using support functions." CAV, 2009.

---

### Official Review · Reviewer_9jBF · 2022-07-12

**Rating:** 7
**Confidence:** 3
**Soundness:** 4 excellent
**Presentation:** 4 excellent
**Contribution:** 3 good

**Summary:**

This work trains a NN (relu) to model a system dynamic of an ODE, then transform this NN into a hybrid automata with affine dynamics. This automata is then verified using dReal, with 2 outcomes. If it is certified, the hybrid automata is a safe abstraction, and is used to prove the safety of the system. If it is not certified, dReal produces a counter example which can be used to refine the NN modeling of the system.

**Questions:**

I have a feeling that, the CEGIS algorithm, when applied to generating examples for numerical optimization (such as training a NN) can have some wild responses. For instance, it is a known problem in polynomial fitting that one should provide sample points in a particular way (https://en.wikipedia.org/wiki/Chebyshev_nodes).

I'm unsure if similar phenonium exists in this domain, but nonetheless it should be reported, as I think it provides insights on the nature of the problem. I propose the following experiment, let me know if it sounds reasonable:

- instead of returning 1 sample point from cegis using dReal, try to get 2 distinct points from dReal (this itself can be done in a fairly hacky way, doesn't matter how).
- randomly choose one of these two points as your counter example
- run until termination, note down properties such as total time spent until a solution is found

repeat the above procedure 100 times. is there a wide distribution difference between solution time? Is there some property of the series of CE points that cause your algorithm to find a safe abstraction sooner?

**Limitations:**

this is done well.

**Strengths And Weaknesses:**

## str

this paper is clean to read, and presents a very straight forward algorithm that simultaneously provides safety guarantees (automatically) while achieving this with as few computation cycles as possible using CEGIS. I'm not too familiar with this field, but if the claims of novelty is substantiated (maybe other reviewers can help me), then I'm in favor of this paper both in soundness and novelty.

## weakness

This paper suffers the classical problem of scalability, which is ubiquitous in thm proving. while we cannot expect this paper to solve the fundamental problem that's the curse of dimensionality, this work should at least discuss _how_ such a problem can be alleviated. As it stands now this work is bit of a "dead end" where it is difficult to see if the proposed algorithm can be extended to become even more efficient.

---

> ### Author Response · Authors · 2022-08-02
> **Response to 9jBf**
>
> We performed this experiment on your advice. At each CEGIS iteration, we sampled two counterexamples by excluding a region around the first, then we chose one randomly. We found no significant difference in total runtime with respect to our previous results. This is likely due to the following two reasons. First, when a counterexample is found, we sample nearby to augment the information the learner gains from the certifier. While these may not all be counterexamples, this ensures a broader coverage of distant points by the learner. Secondly, our sample set initially contains a large number of data points uniformly sampled over the state space to train the network. Counterexamples then do not provide the learner with entirely new regions of the domain, but instead indicate specific areas which normally are few in number.
>
> Verification generally suffers from the curse of dimensionality, though we argue that this limit opens new opportunities for research. Our approach proposes using neural ODEs as abstractions. While our presented method relies on a straightforward reduction to hybrid automata, in future work we see at least two possible directions: (1) simplifying the networks before verification, and (2) designing bespoke reachability procedures for neural ODEs. In this paper, we demonstrated how the expressive power of neural ODEs can enable the verification of systems that are out of reach for existing tools. Verification of neural ODEs is an emergent technology with an active research community. For example, see [3, 4] for the case without disturbance. We expect progress also for neural ODEs with disturbance in the near future.
>
> [3] D. M. Lopez, P. Musau, N. Hamilton, and T. T. Johnson, ‘Reachability Analysis of a General Class of Neural Ordinary Differential Equations’. FORMATS 2022, Available: http://arxiv.org/abs/2207.06531.
>
> [4] S. Gruenbacher et al., ‘GoTube: Scalable Stochastic Verification of Continuous-Depth Models’, AAAI 2022, Available: http://arxiv.org/abs/2107.08467

---

> > ### Comment · Reviewer_9jBF · 2022-08-04
> > **thanks for running the experiments**
> >
> > It would seem that this part "Secondly, our sample set initially contains a large number of data points uniformly sampled over the state space to train the network" is fairly important.
> >
> > Does this large number of data points cause an issue for your solver? For instance making it run slower? Specifically, does your solver need to compute over these sampled data points, or does it only look at a few, localized data points? If the solver has to perform computation over all the sampled data points, there are some literature on selecting a "coreset" of representative points, which might help with your scalability issues:
> >
> > https://proceedings.neurips.cc/paper/2019/file/7bec7e63a493e2d61891b1e4051ef75a-Paper.pdf
> >
> > http://proceedings.mlr.press/v80/pu18b/pu18b.pdf
> >
> > Reading some of other reviewer's responses, it seems scalability and working on a "real" problem is the main complaints. I'd focus on perhaps motivating these ODEs as important, or perhaps even frame it something as "it took scientists months to come up with a provable solution, yet we can automatically do problems like these". This way scalability is less of an issue, because when you're solving real, scientific problems, it is often OK to have a solver run for a month, as long as you get a solution.

---

> > > ### Author Response · Authors · 2022-08-04
> > > **Response to 9jBF**
> > >
> > > Our SMT solver is not affected by the number of data points sampled in the first place. This is because the points are not parts of the input of the certification problem. Our CEGIS loop works in two stages. First, the sampled points are given to the learning component, which uses stochastic gradient descent (PyTorch) and scales to a very large number of points. This produces a trained neural network. Then, this trained neural network and the ODE are passed to the certification component—the SMT solver dReal—which confirms or disproves its validity using symbolic reasoning.
> > >
> > > The only input to the SMT solver are the trained neural network and the ODE. Thus, the performance of the SMT solver is necessarily constant w.r.t. the number of samples and thus constant across CEGIS iterations. Notably, SMT solving takes the majority of computation time in our workflow (see reply to 93re). It being constant w.r.t. sample size supports that improvement in SMT for neural networks (an active field [10]) will lead to a not only large, but also robust performance increase in our method. With today’s technologies we demonstrated the superiority of our method on simple non-Lipschitz systems, which are already out of reach for state of the art verification tools. Obtaining scalability to large systems is a line of work we will pursue in the future.
> > >
> > > [10] Bak, Stanley, et at. "The second international verification of neural networks competition (VNN-COMP 2021): Summary and results." Available: https://arxiv.org/abs/2109.00498

---

### Official Review · Reviewer_Cxvo · 2022-07-16

**Rating:** 6
**Confidence:** 4
**Soundness:** 3 good
**Presentation:** 3 good
**Contribution:** 2 fair

**Summary:**

This paper builds the connection between white-box, non-linear dynamic models to black-box, neural dynamic models. Furthermore, this paper expresses the potential usage of neural abstraction of safety verification tasks, especially for dynamics without local Lipschitz continuity. How to build neural abstractions and verify over the abstractions might be a potentially exciting direction if the scalability issues and details in the paper could be further elaborated on and explored.


—— Updated the score. Thanks for the explanation for the scalability issues. I assume that the authors included the experiment details in the response to me and 93rE in the updated version of the paper.

**Questions:**

- “Corollary 1 ensures that concluding that a neural abstraction is safe suffices to assert that the original dynamical model is also safe.” Does the neural abstraction come with over approximations? If yes, what is the best over-approximation error this neural abstraction can give?
- Similar to Weaknesses 2, what is the justification for the existence assumption of the neural abstraction?
This paper uses $\epsilon$ and $\delta$ to measure the disturbance and the abstraction error. Is there a boundary for these two values? Are they arbitrary customized? What is the minimum $\epsilon$ or $\delta$ one can get?
- The benefits of the synthesis part are not clear. Does the classic regression training method fail to train a neural network for verification? What are the differences in the results from the final verification process between these two training methods? (That said, if the training method does not matter, the verification heavily depends on the setting of $\epsilon$ and $\delta$, which also raises questions in the previous point.)


**Limitations:**

This paper presents an interesting idea without detailed theoretical analysis and experimental evaluation. The approximation and error-bound analysis are expected. Also, more experimental results about the scalability and justification of the current benchmark setting are expected.


**Strengths And Weaknesses:**

*Strengths*

- The idea of replacing non-linear dynamics with neural networks is interesting and potentially useful in the verification community.
- This paper presents better verification results compared to a well-known verification tool.

*Weaknesses*

This paper’s contribution is a bit vague (I think the main contribution is on the verification side, but the author also states the training part as one of the critical contributions. Please correct if my understanding is wrong.) The theorem in the paper comes with the assumption that the neural abstraction exists (The paper leverages a particular category of neural networks: feed-forward neural network with ReLU activations) and lacks existence and convergence guarantee.
-  The main differences between the training method in the paper and the classic regression training process are not well-stated. Using the counterexamples and generating more training samples would guide the search, but how those newly added samples can help with the performance at convergence is not exhibited.
   - One potential experiment: checking if using a large dataset can give a similar performance to using counterexample-based guidance. (The dataset details are not provided in the paper, which is expected.)
- Theorem 1 in the paper assumes a neural abstraction for a dynamical system F exists, which is a key and strong assumption for the following analysis in the paper. What is the reasonable explanation for this assumption? Can the authors provide theoretical analysis for the existence of the neural abstraction given a dynamic F?
- The benchmarks with better performance in neural abstractions are small-scale and not commonly used. The results from these benchmarks are not convincing enough without further details. (e.g., potentially using cases of these benchmarks, further details of the design choices and reasons for these benchmarks, etc.)

---

> ### Author Response · Authors · 2022-08-02
> **Response to Cxvo**
>
> This paper argues that if a neural network satisfies Eq. 4 then the resulting neural abstraction, i.e., Eq. 5, is an over-approximation of the original system (Cor. 1). When Eq. 4 is formally checked using an SMT solver, we guarantee that safety verification over the neural abstraction gives a sound result. Notably, our approach does not guarantee completeness, that is, it may theoretically not find an abstraction. However, we demonstrate empirically that our approach effectively finds abstractions in practice and even enables the verification of non-Lipschitz models. We remark that systems that do not exhibit non-Lipschitz continuity are abundant across literature and encompass engineering and natural phenomena such as in fluid dynamics [1] and the n-body problem [2]. Our results make a step towards an effective verification tool for systems of this kind, which are rarely considered in verification literature.
>
> Q1. A neural abstraction ensures soundness w.r.t. the safety verification question (Cor. 1) because a neural abstraction is an over-approximation of the original model. In fact, neural abstraction is defined as a neural ODE plus a bounded non-deterministic disturbance (Eq. 5). This non-determinism produces a ‘cone’ of solutions that necessarily contains all solutions of the original model. Symbolic safety verification over the induced hybrid automaton handles non-deterministic disturbance. Therefore, we ultimately verify over an over-approximation of the original model.
>
> Q2. Neural abstractions are defined as neural networks that approximate the system up to desired accuracy (Eq. 4) plus a non-deterministic disturbance (Eq. 5). Thus, existence of networks that approximate up to desired accuracy boils down to universal function approximation, which has been shown in seminal work. Giving sufficient conditions for their existence (of a good approximation) is thus a general question in machine learning, out of the scope of our paper. Our message is that given a good approximation one can construct a good—and sound—abstraction for safety verification.
>
> Q3. Formal synthesis augments regression training into “abstraction training” with soundness guarantees. In fact, our learning phase trains a regressor up to the given accuracy over a set of samples. To guarantee formal soundness, our certification phase checks symbolically (using SMT) that the accuracy is guaranteed over the entire domain—notably, our domain is dense and cannot be sampled exhaustively. If certification finds a counterexample, then this is given back to regression training; this results in a CEGIS loop. If certification succeeds, then as our abstraction is sound, it can be given to safety verification. Regression training is thus at the core of our procedure but, alone, it would be insufficient to ensure soundness of the abstraction.
>
> [1] G. L. Eyink and T. D. Drivas, ‘Spontaneous Stochasticity and Anomalous Dissipation for Burgers Equation’, J Stat Phys, vol. 158, no. 2, pp. 386–432, Jan. 2015, doi: 10.1007/s10955-014-1135-3.
>
> [2] F. Diacu, ‘The solution of then-body problem’, The Mathematical Intelligencer, vol. 18, no. 3, pp. 66–70, Jun. 1996, doi: 10.1007/BF03024313.

---

> > ### Comment · Reviewer_Cxvo · 2022-08-04
> > **Thanks for the response.**
> >
> > Thanks for the authors' response.
> >
> > `Giving sufficient conditions for their existence (of a good approximation) is thus a general question in machine learning, out of the scope of our paper. Our message is that given a good approximation one can construct a good—and sound—abstraction for safety verification.` This part is fair to me. But I am still concerned about the lack of detailed scalability evaluation, which would be of interest to understand the scope of this work.
> >
> > While reading the other reviewers' replies and the updated supplementary materials, two new questions came up:
> > - In Appendix B, you mentioned `We note that since ASM-based abstractions are constructive and are able to deterministically increase the number partitions and consequently reduce the error, for very large numbers of partitions they would achieve lower errors than neural abstractions. However, in practice these abstractions would be too large in complexity to use with SpaceEx for safety verification.` Would you elaborate on the `very large numbers of partitions` here? As in Table 1, $N_p$ for Jet Engine in Affin Simplicial Mesh is 128, which already achieves a 0.083 error bound. Can $N_p$ == 256 or 512 achieve lower or comparable error bound? 256 or 512 does not seem to be super large partitions.
> > - As a follow-up to the first question and the comments from 93rE, adding other verification tools, e.g. Flow*, inside the automatons analysis seems necessary for a better understanding of the performance position of this work. One Flow* version [1] also states that Flow*1.2 is more effective in playing with hybrid systems than SpaceEx.
> >
> > [1] Chen, X., Sankaranarayanan, S., & Abrahám, E. (2015, December). Flow* 1.2: More Effective to Play with Hybrid Systems. In ARCH@ CPSWeek (pp. 152-159).

---

> > > ### Author Response · Authors · 2022-08-08
> > > **Response to Cxvo**
> > >
> > > Q1. Verification tools for hybrid automata are designed for systems with a number of modes in the order tenths. Notably, no model in the aforementioned ARCH competition [5] has more than 100 modes. The purpose of Table 1 in Appendix B is to show that neural abstractions achieve a given error for significantly fewer modes than naive grid partitioning. This allows us to effectively exploit hybrid systems verifiers, in particular SpaceEx. In fact, at 200 partitions, the error for affine simplicial mesh would be 0.053; in contrast neural abstractions achieve on average an error of 0.058 for just an average of 68 partitions. Our goal is to have as few modes as possible for the same error, which neural abstraction enables us to do.
> > >
> > > Q2. The cited paper does not claim to be more effective the SpaceEx, but instead more effective than Flow* 1.0. Indeed, it concludes ‘The performance on linear systems is greatly improved and comparable to SpaceEx’. Flow* is thus at most as good as SpaceEx on their experiments on linear systems. We emphasise that this enhances our argument for using SpaceEx, which is an efficient and specialised tool for the linear case. We made a further experiment to confirm this fact; we further elaborate on this in the reply to 93rE.

---

> > > > ### Comment · Reviewer_Cxvo · 2022-08-08
> > > > **Thanks for the response.**
> > > >
> > > > Thanks for the explanation and additional experiments. I will update my score.

---

### Author Response · Authors · 2022-08-02
**Overall Response**

We thank the reviewers for their comments and questions. In the final version, we will integrate their suggestions and clarify all points made in this rebuttal.

---

### Meta-Review · Area_Chair_r5vS · 2022-08-25

**Recommendation:** Accept
**Confidence:** Less certain

**Metareview:**

This was a borderline paper. The overall idea of using neural networks as abstractions is sound and novel. However, the evaluation is not thorough, there is no theoretical analysis, and concerns about scalability remain. After calibrating across my pile, I am recommending acceptance. Please make sure to incorporate the reviewers' feedback in the final version of the paper.

**Award:**

No

---

### Decision · Program_Chairs · 2022-09-14

Accept